# Calorimetric Markers for Detection and Monitoring of Multiple Myeloma

**DOI:** 10.3390/cancers14163884

**Published:** 2022-08-11

**Authors:** Sashka Krumova, Svetla Todinova, Stefka G. Taneva

**Affiliations:** Institute of Biophysics and Biomedical Engineering, Bulgarian Academy of Sciences, Acad. G. Bonchev Str. Bl 21, 1113 Sofia, Bulgaria

**Keywords:** differential scanning calorimetry, multiple myeloma, monoclonal proteins, free light chains, monoclonal gammopathy of undetermined significance, Waldenström macroglobulinemia, transition temperature, excess heat capacity, albumin, immunoglobulins

## Abstract

**Simple Summary:**

This review highlights the potential of differential scanning calorimetry for multiple myeloma diagnosis and monitoring of the treatment outcome. The thermodynamic signatures of blood sera from patients with multiple myeloma are strongly dependent on the concentration and isotype of the secreted monoclonal immunoglobulins. Mathematical methods developed to analyze the biocalorimetry data and distinguish “diseased” from “healthy” thermogram to stratify plasma calorimetric profiles and determine specific interrelations between calorimetric and biochemical/clinical data are discussed.

**Abstract:**

This review summarizes data obtained thus far on the application of differential scanning calorimetry (DSC) for the analysis of blood sera from patients diagnosed with multiple myeloma (MM) with the secretion of the most common isotypes of monoclonal proteins (M-proteins), free light chains (FLC) and non-secretory MM, as well as Waldenström macroglobulinemia and the premalignant state monoclonal gammopathy of undetermined significance. The heterogeneous nature of MM is reflected in the thermal stability profiles of the blood serum proteome of MM patients found to depend on both the level and the isotype of the secreted M-proteins or FLC. Common calorimetric markers feature the vast majority of the different myeloma types, i.e., stabilization of the major serum proteins and decrease in the albumin/globulin heat capacity ratio. A unique calorimetric fingerprint of FLC molecules forming amorphous aggregates is the low-temperature transition centered at 57 °C for a calorimetric set of FLC MM and at 46–47 °C for a single FLC MM case for which larger aggregates were formed. The calorimetric assay proved particularly advantageous for non-secretory MM and is thus a suitable tool for monitoring such patients during treatment courses. Thus, DSC provides a promising blood-based approach as a complementary tool for MM detection and monitoring.

## 1. Application of Differential Scanning Calorimetry for Plasma/Serum Proteome Characterization

Differential scanning calorimetry (DSC) has rapidly evolved as a tool for diagnosis of a variety of diseases including cancer, autoimmune diseases, diabetes, etc., as outlined in the pilot reports of Garbett et al. [1,2]. DSC holds great potential for application in routine clinical practice since the measurements are fast and cheap, the plasma/sera are derived in a minimally invasive way, and an automated sampling system is already commercially available. Furthermore, blood plasma/serum samples were shown to be very stable—their calorimetric characteristics are hardly changed upon storage in a refrigerator for a week or at −20 °C for 6 months [3], which allows investigations of samples stored in biobanks.

This biophysical technique is classically applied to study and thermodynamically describe the stability and conformational transitions of biomolecules in solution. However, DSC is also suitable to analyze complex biofluids such as blood plasma/serum [3,4,5], cerebrospinal fluid [6,7], and synovial fluid [8], the calorimetric profiles (thermograms or scans) of which are a complex mixture of the thermal transitions of the individual biofluid components. DSC profiles depend not only on the plasma/serum proteins content and conformation but also on factors that moderate their stability such as ligand binding, macromolecular complex formation, mutations, chemical changes, protein oligomerization, aggregation and misfolding.

For a multicomponent system such as blood serum, a number of overlapping transitions is commonly observed. The average DSC profile, i.e., excess heat capacity (c_P_^ex^) vs. temperature curve, recorded for healthy sera (controls), is presented in Figure 1; it is composed of three well-resolved transitions at 62 °C, 68 °C and 86 °C and a shoulder at 75 °C (Figure 1). The nature of those peaks was deduced taking into account the DSC scans of isolated plasma proteins and their relative abundance in plasma/serum [3,4,9]: the major transition is assigned to the most abundant serum protein albumin (it constitutes about 55% of the total serum protein content in healthy individuals [10]), the peak at 68 °C and the shoulder at 75 °C are dominated by immunoglobulins, and the highest-temperature transition at 86 °C is due to both IgG and transferrin (Tf). Blood plasma features one more transition at ca. 51 °C that is assigned to fibrinogen [4,9]. The thermodynamic parameters of unfolding of the major (albumin and immunoglobulins) serum proteins derived from the calorimetric profiles of “healthy” and “diseased” sera as well as the statistical parameters commented in this work are summarized in Table 1.

The position and amplitude of the resolved transitions as well as the weighted average center (T_FM_) of plasma/sera thermograms were shown to be rather constant for serum derived from healthy individuals [1,2,4]. These thermodynamic parameters were shown to change significantly for a number of diseases [1,4,9,11,12,13,14,15,16,17,18,19], and especially for various types of cancer—breast [20,21,22,23], pancreatic [24], cervical [25,26,27], lung [28,29,30,31,32], brain [6,31,33], colorectal [18], gastric adenocarcinoma [34], melanoma [35,36,37], multiple myeloma (MM) [5,38,39,40,41,42,43], chronic lymphocytic leukemia and acute myeloid leukemia [44]—and can therefore be used to stratify DSC data into different disease-related groups.

Different mathematical approaches were developed for additional evaluation of the extent to which a “diseased” thermogram differs from a “healthy” one, the first one being the algorithm developed by Fish at al. [45] that compares a test (“diseased”) thermogram to a set of control thermograms. As an outcome, it derives the following statistical parameters: r, Pearson’s correlation coefficient, which evaluates differences in the shapes of the two curves; P, spatial distance metric, which reflects the deviation in the distance between the two curves at each temperature point; and ρ, similarity metric, which combines r and P (Table 1). Values close to 1 indicate high similarity of the test thermogram and the control set, while values deviating from 1 are due to statistically significant differences between the two curves [45]. Later on, a number of parametric and non-parametric tests were also applied. Rai et al. [26] developed a parametric method and proved its high efficiency for DSC plasma thermogram stratification; they compared the parametric with a non-parametric and semi-parametric approach for modeling high-dimensional datasets for cancer classification [27]. Garbett et al. [46] utilized modified linear discriminant analysis for the classification of lupus patients that, combined with serological marker information, increased the sensitivity and overall accuracy of patient diagnoses. Multiparametric deconvolution analysis helped discriminate and differentiate the stages of gastric adenocarcinoma [34] and detect and even foresee melanoma before its clinical manifestation [37]. Our team employed the InterCriteria Analysis (ICA) concept (originally developed by Atanassov [47]) and compared it to Pearson’s and Spearman’s correlation analyses of datasets combining calorimetric, statistical, and biochemical data collected for multiple myeloma and colorectal cancer patients [48,49]. This approach revealed correlations between the calorimetric and biochemical parameters of blood plasma/serum, which helped in deciphering the nature of the shifts in the thermal stability of the major serum proteins. Our data showed that these correlation-based methods are complementary for the revelation of complex interrelations between the tested criteria and for stratification of the disease thermograms [48,49].

## 2. Multiple Myeloma Discrimination and Calorimetry-Based Classification

Our pilot calorimetric investigation on MM started with a heterogeneous population of patients diagnosed with MM with the secretion of different isotypes of paraproteins (M-proteins)—IgG, IgM, IgA, as well as cases with secretion of free light chains (FLC) and non-secretory (NS) MM [5]. A large variation in the DSC profiles of blood sera dependent on the MM type (secretory/non-secretory) and paraprotein concentration was found. To explore the significance of the MM type for serum calorimetric characteristics, a discrete large population for each type was examined over the years. The calorimetric profiles of MM sera for each MM type were stratified in several sets on the basis of the number of the successive thermal transitions, their T_m_ and amplitude [5,38-42].

Here, we review the entire database of DSC profiles derived for MM in order to: (i) define general calorimetric MM classification, (ii) identify typical DSC signatures for the secretory and non-secretory MM types, (iii) compare them with those of the premalignant condition monoclonal gammopathy of undetermined significance (MGUS) and other monoclonal gammopathies and (iv) critically evaluate the potential of DSC for MM detection and monitoring.

We re-evaluated a database of 643 DSC profiles of MM patients which were either previously grouped in sets of thermograms for each MM type [5,38,39,40,41,42] or were newly recorded. Here, the thermograms are combined in four groups irrespective of the different myeloma types (IgG, IgM, IgA, FLC and NS MM) based on the number of the thermal transitions, the position of the main transition and the similarity in shape of the DSC profiles (Figure 2a,b). Two IgG MM (IgG1 and IgG2) and two FLC MM (FLC1 and FLC2) sets could not be grouped due to their unique calorimetric features (Figure 2c,d).

As can be seen in Figure 3, FLC MM and NS MM cases contribute most to the MM1 set (63% and 83% of the FLC MM and NS MM cases, respectively), but it also contains 24–39% of all other MM types. That is the group of thermograms that most resembles the control set, T_m_^HSA^ and T_m_^Igs^ are close to the control ones but the c_P_^HSA^/c_P_^Igs^ ratio is lower than that of the control and the last transition that occurs at 86 °C in controls is missing in most cases; T_m_^HSA^, T_FM_ and the statistical measures (r, P and ρ) deviate significantly from the control values (Table 2). The rest of NS and FLC MM cases are part of set MM2, along with 24% of the IgG MM cases. For this set, a significant fraction of the HSA molecules denatures at higher temperature than the control as evidenced from the up-shifted T_m_^HSA^ transition to 66 °C but the c_P_^HSA^/c_P_^Igs^ ratio remains > 1; T_FM_, r, P and ρ deviate further from the control values (Table 2). Both MM3 and MM4 sets exhibit two transitions with low c_P_^HSA^ and high c_P_^Igs^ that results in a c_P_^HSA^/c_P_^Igs^ ratio much lower than 1. Together, these two groups encompass 76% of all IgA MM cases, while group MM4 is largely dominated by IgM MM (75% of the IgM MM cases). For these two sets of thermograms, T_FM_ has the highest, while r, P and ρ have the lowest values among the four sets (Table 2). The enthalpy ΔH_cal_ for the four calorimetric sets is slightly larger than that of the control set (Table 2). Two of the defined IgG (only) MM sets (IgG1 and IgG2, accounting for 16% of all IgG MM cases) exhibit specific features that distinguish them from all other MM sets, i.e., high amplitude transitions at 67 °C and 75 °C, and well-resolved transition at 82 °C originating from Tf and IgG (in contrast to the minor transition at 86 °C in control thermograms). The denaturation temperature of the main transition has the highest value for IgG1 set among all defined MM sets. For IgG1 and IgG2 sets the albumin transition is largely shifted to high temperatures and significantly overlaps with the globulins’ transitions, which obscures their clear distinction.

A total of 23% of the FLC cases also exhibit specific features, i.e., high-amplitude sharp transition at 61 °C (set FLC1) or low amplitude transition at 57 °C (set FLC2). While the nature of the 61 °C transition is not clear yet, the origin of the 57 °C peak can be ascribed to FLC denaturation considering: (i) the high correlation between the amplitude of this transition and FLC serum level, (ii) the close coincidence of the transition temperature to that of the denaturation of isolated FLC molecules [50], and (iii) significant reversibility of the denaturation proved by us applying annealing procedure [39] and typical for free FLC [50,51,52].

For a single FLC MM case (stage III, International Staging System (ISS) classification) with κ FLC concentration above 20% the 57 °C transition was found to be shifted to 46 °C and attributed to destabilized FLC molecules [39]. This calorimetric feature was preserved for a monitoring period of 12–25 months after diagnosis and was not reported for any other MM case or other disease.

In contrast to control sera where no aggregation was evident in the atomic force microscopy (AFM) images, these thermally unstable FLC were shown to form aggregates with amorphous structure and height above 7.5 nm (Figure 4). Much smaller (ca. 4 nm) aggregates were also observed for FLC case included in the FLC2 calorimetric group (Figure 4, [39]).

To determine the extent to which the observed changes in the amplitudes of the calorimetric transitions are due to changes in the level of the major protein constituents of the blood serum we applied serum protein electrophoresis (SPE). This technique provides the concentration of albumin, α-, β- and γ-globulin fractions, and resolves the presence of M-protein as a clear spike either in the β- or γ-globulins region [53]. The accumulated calorimetric and SPE data were included in a database that was subjected to correlation and ICA analyses. This allowed us to reveal complex interrelations between the calorimetric features and the protein composition of MM blood sera.

The correlation analyses provide numerical expression for the correlation between pairs of parameters that are linearly dependent. The ICA method on the other hand searches for consonances between multiple parameters by comparing the behavior of the input objects (predefined groups of MM cases) against the input criteria (calorimetric and SPE data). It results in the evaluation of the relations (<, =, >), and not their numerical expression. The index matrix relating the objects (in our case, MM patients) and the criteria (the blood indicators, i.e., the thermodynamic and SPE parameters) are then transformed into an index matrix that provides the degree of similarity (μ) and differentiation (ν) between pairs of criteria, i.e., intercriteria correlations in the [0;1] interval. For the purpose of our study we defined the following threshold values: μ(ci,cj) > 0.75 and ν(ci,cj) < 0.25 for strong positive consonance, μ(ci,cj) > 0.70 and ν(ci,cj) < 0.30 for weak positive consonance, μ(ci,cj) < 0.25 and ν(ci,cj) > 0.75 for strong negative consonance, μ(ci,cj) < 0.30 and ν(ci,cj) > 0.70 for weak negative consonance; intermediate values were regarded as dissonance/uncertainty (for more details see Krumova et al. [49]).

We have established that a common feature between all calorimetric MM sets is the lower amplitude of the albumin assigned thermal transition at 62 °C as compared to healthy controls, an effect that was independent of the albumin concentration (dissonance between albumin concentration and c_P_^HSA^, Table 3). It also became clear that the amplitude of the Igs assigned transition is predominantly determined by the M-protein concentration (strong consonance between c_P_^Igs^ and M-protein concentration, Table 3); the increase in c_P_^Igs^ is also associated with decrease in c_P_^HSA^ (strong negative consonance for this pair of criteria, Table 3), which confirms the stabilization of the albumin molecules and overlap of their thermal transition with the one of M-proteins.

These analyses also revealed that the position of the Igs peak depends on the type of M-protein, i.e., when it migrates in the β fraction of the SPE profile, T_m_^Igs^ is shifted to temperatures up to 76 °C, while when it migrates in the γ-proteins region, T_m_^Igs^ is not shifted. The heat capacity of globulin transition, c_P_^Igs^, is also in strong consonance with both the weighted average center of the thermograms, T_FM_, and the statistical measure ρ, and thus, it is the major factor that affects them [49].

## 3. Calorimetric Signatures of Secretory and Non-Secretory MM Types

It can be stated that from the generated calorimetric sets (Figure 2), the MM1 set is the most typical for FLC and NS MM cases, set MM4 is representative for IgM MM cases, while most of the IgA MM cases are spread among sets MM3 and MM4. IgG MM cases are the most heterogeneous (distributed across MM1–3 sets as well as in two specific sets of profiles). Nearly a quarter of the FLC cases also show specific calorimetric features, i.e., transition at 57 °C that was hypothesized and proved to originate from the denaturation of FLC, while a fraction of the FLC was suggested to form complexes with other molecules and to melt at higher temperatures overlapping with the globulins’ transitions (sets FLC1 and FLC2) [38]. Among all reported DSC scans of sera, only one FLC MM case exhibited a unique transition located at 46–47 °C (Figure 2d). This unique transition was fully reversible and was assigned to the denaturation of destabilized FLC molecules [39].

Special attention must be paid to the application of DSC as a tool for the detection/diagnosis and monitoring of the subset of MM which do not produce intact monoclonal immunoglobulins or FLC detectable by SPE, i.e., NS MM. Due to the lack of serum biomarkers, the diagnostics and monitoring of those cases are more difficult since expensive and/or invasive approaches must be used. With the development of the more sensitive FLC assay [54,55] that can detect small amount of monoclonal FLC in sera, these problems are overcome to a large extent and this method becomes routine for NS MM monitoring. Nevertheless, the problem remains for the so-called true non-secretors, which suffer from end organ damage but no serum monoclonal proteins are produced; therefore, a skeletal survey and marrow plasmacytosis must be performed in order to evaluate the disease progression. Since NS MM cases are rare (3–5%) and true NS accounts for only 1% of all MM cases [56,57], and due to the difficulties associated with monitoring of the response to treatment of such patients, they are generally excluded from clinical trials. To gain insight into the nature and course of progression of NS MM in our previous study, we have evaluated the DSC scans of 201 such cases for which 132 had a κ/λ FLC ratio within the reference limits and therefore can be regarded as true non-secretors, while 59 had an abnormal κ/λ FLC ratio (oligo-secretory MM) and for 10 cases this parameter could not be determined due to limitations of the method [38]. All those cases are part of the calorimetric groups MM1 and MM2, as defined above, and hence, their thermograms deviate from the control ones to a lesser extent than the other groups dominated by secretory MM cases, which is probably associated with the better prognosis of NS than of secretory MM cases. Our data strongly suggest that DSC has potential as a complementary tool for monitoring the response to treatment of those patients in a fast and non-invasive way. To this end, it is not clear what the trigger of the deviation of thermograms from control values is since even for true non-secretors, the DSC profiles are altered that should be due to modified protein composition and/or binding state (protein–ligand, protein–protein interactions) in the blood serum of those patients [58].

It should also be noted that the DSC scans of NS MM samples from the MM1 set are similar to the ones reported for diabetes type I patients with early renal function decline [11] that also exhibit a decrease in the amplitude of the 63 °C transition and increase in that of the 70 °C peak. One of the reported cases in the work of Garbett et al. [11] also resembles the MM2 set. Therefore, a systematic study of the relationship between the calorimetric features of NS MM as well as the other MM types and the renal function is of a high interest. The role of other symptoms such as hypercalcemia, anemia and the presence of bone lesions also must be thoroughly evaluated.

So far, our results show that the most sensitive calorimetric parameter for NS MM detection is the abnormal T_FM_; the combination of the immunological marker κ/λ FLC ratio and the calorimetric parameter T_FM_ significantly increases the sensitivity and specificity of NS detection [40].

## 4. Monitoring of MM Patients by DSC

Blood sera from 11 patients (10 with secretory MM, isotypes IgG, IgA, IgM, and 1 case with FLC MM) were derived directly before autologous stem-cell transplantation (ASCT) and during a period of 12–16 months after the transplantation [43]. In addition, samples from patients with secretory MM, FLC MM, NS MM and Waldenström’s macroglobulinemia were also obtained before the start and during the treatment course (up to 20 months) [40]. This allowed us to follow the changes that occur in the respective DSC profiles during the course of treatment of the disease, to correlate them with the levels of immunological markers and to judge whether the DSC approach accurately reflects the patients’ clinical status; selected examples are presented in Figure 5.

The changes in the calorimetric parameters were compared with the variation in the M-protein concentration and the κ/λ FLC ratio [43]. A very individual response to the treatment was observed for both transplanted and non-transplanted patients with respect to the calorimetric features and immunological markers. ASCT led to 50% recovery of the albumin assigned transition to the control value, while this percentage was reduced by half for non-transplant patients. The globulins region in the DSC profile generally did not revert to the control values for both transplanted and non-transplanted individuals. For both types of patients, the parameters T_FM_ and r were found to correlate best with the serum level of M protein (Figure 5), although few examples of originally secretory MM that reverted to NS MM (M-protein was not detectable by SPE) were found to manifest calorimetric features different from the control ones. Nevertheless, the improvement of the clinical status of secretory MM patients was reflected in the DSC profiles. Most of the thermograms of NS MM cases deviated from the control ones, even though no serum biomarkers were detectable, which strongly indicates abnormal molecular interactions that do affect the calorimetric features but remain “invisible” for the standard immunological tests.

In our earlier work, we have determined the specificity and sensitivity of the immunological test (that includes the M-protein concentration and κ/λ FLC ratio) and the calorimetric approach (taking into account the r and T_FM_ parameters) for MM diagnosis [40]. We found that the M-protein level is more sensitive and specific than r and T_FM_; however, for cases in which no M-protein is secreted, the calorimetric markers perform better than the immunological marker κ/λ FLC ratio. The combination of immunological markers and T_FM_ increased the specificity and sensitivity of disease detection, most pronounced for NS MM cases (for further details see [40]). Therefore, it is our belief that DSC might be a useful complementary monitoring tool for MM patients.

## 5. DSC Based Discrimination of MM and Other Diseases

Alike MM patients, a small cohort of patients with MGUS [57], a premalignant state that is known to precede myeloma, exhibited diverse DSC profiles [16]. A number of similar calorimetric features between the two hematological disorders can be identified: (i) IgG MGUS cases exhibit the largest heterogeneity in terms of thermograms shape and position of the main transition; (ii) the shape of some of the IgG MGUS thermograms resembles those included in the MM2 set but have higher enthalpy, while the others are similar to those in sets MM2, MM3 and MM4; (iii) for all MGUS cases, the amplitude of the transition at 62 °C decreases while that of 68–76 °C increases, resulting in significantly higher T_FM_ value compared to the controls. Thus, DSC is not only able to detect the early stage of MM development but also demonstrates that similarly to the later stages of MM, this premalignant disorder is characterized by the stabilization of HSA and M-protein-attributed transitions.

Nearly identical calorimetric scans were recorded for a small cohort of non-MGUS patients with other immunological pathologies/monoclonal gammopathies that, however, were not explicitly specified [16]. A more in-depth study compared the calorimetric features of patients diagnosed with MM and Waldenström’s macroglobulinemia, both pathologies being related to the secretion of monoclonal IgM proteins [41]. It was also established that there is no dramatic difference between the two pathologies with respect to the calorimetric features with the exception of the fact that a larger portion of WM sera than of IgM MM ones falls in the calorimetric set with features that slightly differ from healthy controls, while more IgM MM than WM cases take part in the calorimetric group that deviates the most from the control set. This can probably be attributed to the less aggressive nature of WM than IgM MM.

The next question was whether MM sera can be discriminated from other non-hematological malignancies on the basis of their calorimetric features. A survey of the published data so far reveals that reduction in the amplitude of the albumin assigned thermal transition at 63 °C and its stabilization (a shift by 2–3 °C) is observed not only for MM but also for melanoma [9,36], endometrial cancer, amyotrophic lateral sclerosis, rheumatoid arthritis [9], cervical disease [25,26], stage II breast cancer [22], stage I gastric adenocarcinoma [34], and one case of CLL and AML [44]. In all those works, however, the albumin concentration was not provided, so it is not clear whether the reduction in the albumin assigned transition was due to lower albumin level or not. Even more dramatic up-shift of T_m_^HSA^ accompanied by increase in the amplitude and/or T_m_ of 70 °C or 75 °C transitions was reported for cervical cancer [1], lung cancer, Lyme disease, systemic lupus erythematosus [3,9,46], diabetic patients with CAD+ [2], one patient with diabetes + ERFD [11], stage III breast cancer [22], severe stage of chronic obstructive pulmonary disease [13], breast cancer [20,21], a cohort of schizophrenia patients [15], and non-cancerous inflammation [18]. However, only for MGUS [16], WM [40,41] and MM (see sets MM3, MM4, IgG1, IgG2), and some cases of Lyme disease [9], which is associated with production of IgM and IgG antibodies, the amplitude of the immunoglobulins assigned transition at 70 °C or 75 °C was as high as that of albumin assigned transition at 62 °C in healthy subjects. Thus, high-amplitude Igs transition is most often associated with the presence of monoclonal proteins.

Concerning the stabilization of albumin against the thermal challenge, chemical/conformational modifications and/or altered binding states of the most abundant plasma/serum protein can be suggested [58]. For example, albumin oxidation results in an increase in both T_m_ (by 15 °C) and enthalpy (by 100 kJ/mol) of its temperature-induced unfolding [59]. Therefore, extensive production of ROS species associated with cancer metabolism might be one plausible reason for the change in T_m_^HSA^ in MM as well as other malignancies. In particular, for the case of MM, lowering of the enzymatic and non-enzymatic antioxidants level and increase in oxidative stress markers have been reported [60,61,62]. It has also been demonstrated that binding of small molecules such as fatty acids or the formation of macromolecular complexes between plasma proteins is able to stabilize the albumin transition [63]. Therefore, the trigger for the altered DSC profiles in different diseases remains unclear so far, but it is highly probable that altered intermolecular interaction networks play crucial role for the occurrence of these unspecific thermodynamic features.

The occurrence of a peak at 56–58 °C observed for group FLC2 in our study was also reported for stage III breast cancer [22], and a cohort of schizophrenia patients [15]. However, in the case of FLC MM, it can safely be ascribed to the denaturation of monoclonal FLC, while in the other studies, its nature was not clarified.

## 6. Conclusions

In this work, we reassessed data on the DSC application for the study of blood sera from patients diagnosed with MM with the secretion of the most common isotypes of monoclonal proteins (IgG, IgM, IgA), FLC MM and non-secretory MM, gathered by us over the last decade. The evaluation of the calorimetric and statistical parameters associated with each of the calorimetry-based MM sets of thermograms allowed for the identification of calorimetric signatures of MM: stabilized albumin and immunoglobulins assigned transitions, high amplitude of the 70–76 °C transition due to the accumulation of M-proteins, the occurrence of transitions at 47 °C and 56 °C ascribed to monoclonal FLC with different degree of stability, increased weighted average center of the thermogram (T_FM_) and reduced values of the statistical measures describing the deviation from the healthy thermogram.

Although there are still a number of issues that need to be clarified before implementing DSC in the clinical practice, our data strongly suggest that is can be used as a complementary technique to the currently applied clinical methods for MM diagnosis (Figure 6).

The data strongly suggest that DSC is a suitable novel tool for secretory and non-secretory MM detection and monitoring in a fast, non-invasive, and informative manner. Finally, this review provides important information to bridge the calorimetry assay and clinical tests for the diagnosis of cancer and other diseases and is thus a basis for further application of biocalorimetry to other blood malignancies and uncommon blood cancer types.

## Figures and Tables

**Figure 1 cancers-14-03884-f001:**
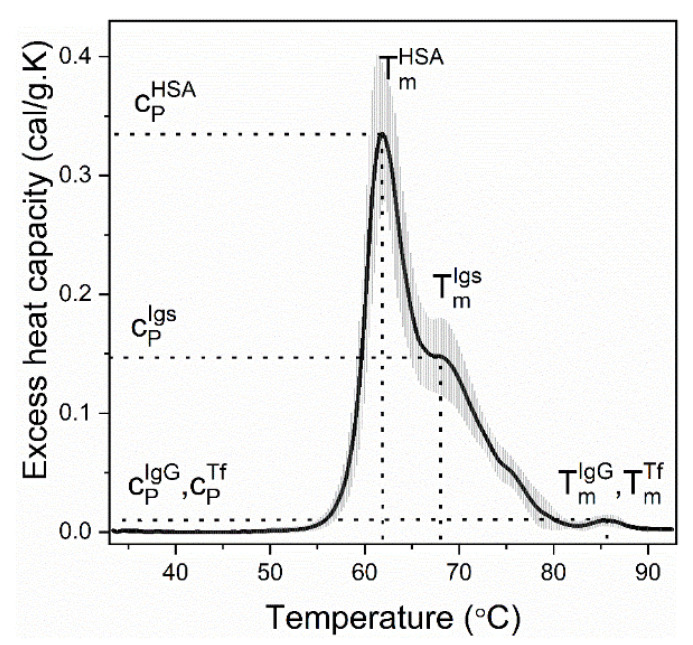
Control set of serum thermograms (mean, black line; SD, grey shadow) derived for healthy individuals. Before analysis, buffer–buffer scans, routinely measured before each experiment, were subtracted from the serum scans; the resulting traces were then corrected either by linear baseline or by interpolating sigmoidal baseline (when the slopes of the pre- and post-transition baseline differed) and normalized to the total protein concentration. The transition temperatures of the successive thermal transitions assigned to the major serum proteins: albumin (T_m_^HSA^ and c_P_^HSA^) and immunoglobulins (T_m_^Igs^ and c_P_^Igs^) and the high temperature transition to transferrin (T_m_^Tf^ and c_P_^Tf^) and IgG (T_m_^IgG^ and c_P_^IgG^) are indicated.

**Figure 2 cancers-14-03884-f002:**
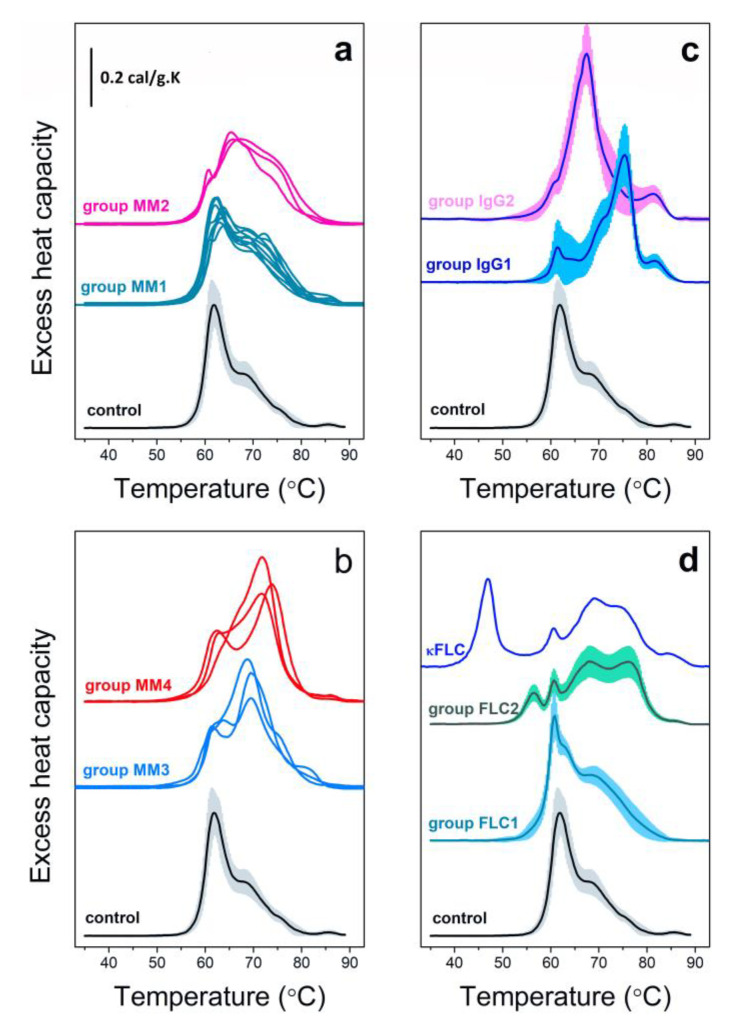
General calorimetric classification of MM cases. Average sets of DSC profiles of MM sera recorded for different MM types (129 cases of IgG [5,37,40], 52 cases of IgM [5,37,38,40], 44 cases of IgA [5,37,39,40], 75 cases of FLC [5,35,36,37,40] and 296 cases NS [5,37,40] MM) are presented; the combination in groups MM1-MM4 is based on number of peaks, similarity in shape and on the position of the main peak (**a**,**b**). For clarity only the average thermograms, without the corresponding SD, are presented. Specific MM groups for 24 IgG MM cases (**c**) and 23 FLC MM cases (**d**) (mean ± SD), as well as unique FLC case differ from the rest of investigated MM cases ((**d**), blue line).

**Figure 3 cancers-14-03884-f003:**
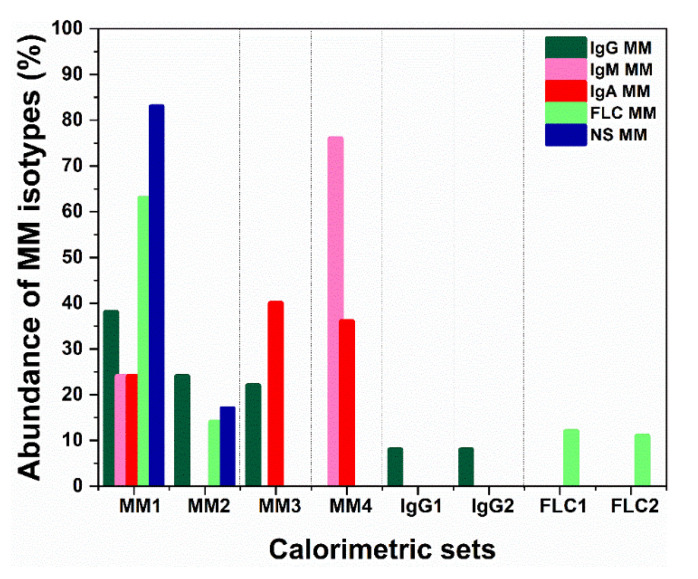
Distribution of each MM isotype among the different calorimetric sets defined in Figure 2. The color-coded bars represent the % of MM cases of certain isotype in each calorimetric set relative to the total number of cases for the same MM isotype.

**Figure 4 cancers-14-03884-f004:**
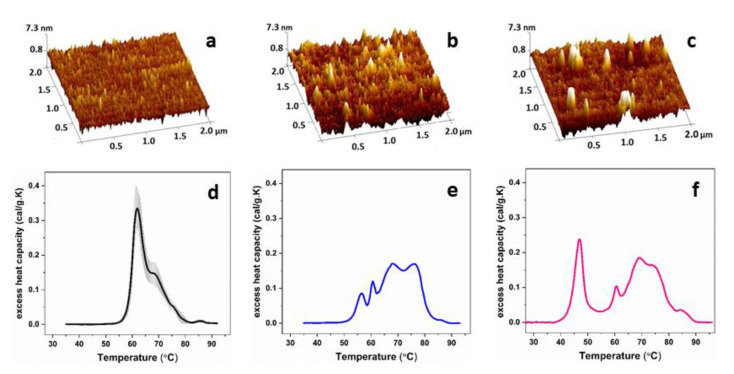
Atomic force microscopy images of blood serum of healthy control (**a**), a case from FLC2 calorimetric group (FLC concentration 10.5 ± 7.8% of the total protein content) (**b**) and a specific κ-FLC MM case (stage ISS III) with FLC concentration above 20% of the total protein content (**c**). The sera are deposited onto mica coverslip and measured in air. The white areas (with height above 4 nm) are due to amorphous aggregates). Calorimetric profiles of blood serum samples (**d**–**f**) corresponding to the AFM images in a–c, respectively. Modified from [39].

**Figure 5 cancers-14-03884-f005:**
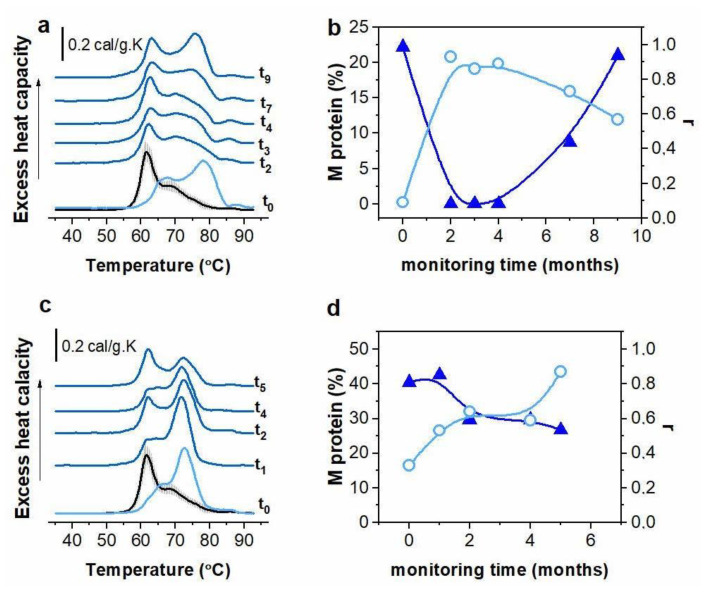
Selected series of calorimetric scans recorded for 2 patients monitored for 5 or 9 months: IgA MM patient before and after ASCT (t_0_—before, t_i_—monitoring points after transplantation, subscripts denote months after ASCT) (**a**) and IgM MM patient during the course of the applied clinical treatment (t_0_—beginning of monitoring period, the months of monitoring period are indicated as t subscript) (**c**). The mean DSC profile for healthy individuals not subjected to medical intervention is plotted in black (SD, grey shadow). Correlation between the variation in the M-protein concentration ((**b**,**d**), blue triangles) and the shape similarity parameter r ((**b**,**d**), light blue open circles) for the monitoring points after transplantation are shown for the two presented cases. Modified from [43].

**Figure 6 cancers-14-03884-f006:**
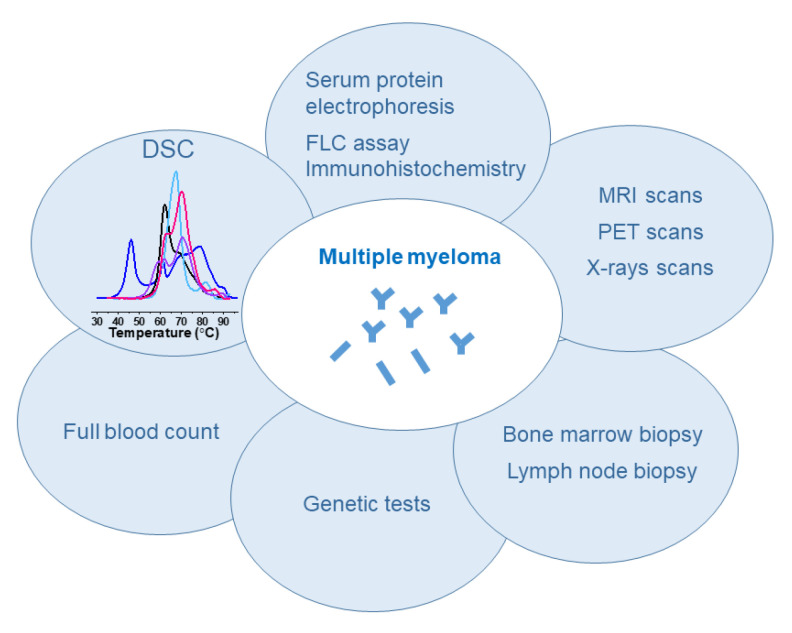
Schematic representation of interdisciplinary approach for MM diagnosis involving DSC as a complementary tool to clinical tests.

**Table 1 cancers-14-03884-t001:** Calorimetric parameters determined from blood serum DSC profiles, statistical measures describing the extent of resemblance between the “diseased” and control “healthy” calorimetric set of thermograms and degree of similarity/differentiation between pairs of criteria (ci, cj) determined for sets of (multiple myeloma) MM thermograms by InterCriteria Analysis.

Calorimetric Parameters	
T_m_*^i^*	transition temperature, *i* is either albumin (HSA) or immunoglobulins (Igs)
c_P_^i^	excess heat capacity, *i* is either albumin (HSA), or immunoglobulins (Igs)
c_P_^HSA^/c_P_^Igs^	ratio of the excess heat capacities of albumin and immunoglobulins transitions
ΔH_cal_ = ∫c_P_. dT	enthalpy of the thermogram (integrated area of the thermogram)
$T_{\mathrm{FM}}=\int_{T_{1}}^{T_{2}}Tc_{P}{}^{ex}\left(T\right)dT/\int_{T_{1}}^{T_{2}}c_{P}{}^{ex}\left(T\right)dT$ where *T1* is the initial and *T2* is the final temperature point of the thermogram	weighted average center of the thermogram
Statistical Measures	
r	Pearson’s correlation coefficient
P	spatial distance metric
$\rho ={\left(P^{w}r^{2-w}\right)}^{1/2}$	similarity metric
where 0 *≤ w ≤* 2	
InterCriteria Analysis	
μ(c_i_,c_j_)	degree of similarity between the pair of criteria (c_i_, c_j_) determined for sets of MM thermograms by InterCriteria Analysis
ν(c_i_,c_j_)	degree of differentiation between the pair of criteria (c_i_, c_j_) determined for sets of MM thermograms by InterCriteria Analysis

**Table 2 cancers-14-03884-t002:** Values of the calorimetric parameters T_m_^HSA^, T_m_^Igs^, c_P_^HSA^/^Igs^, T_FM_, ΔH_cal_, and the statistical parameters Pearson’s correlation coefficient (r), spatial distance metric (P) and similarity metric parameter (ρ) derived from comparison of the MM thermograms included in each of the calorimetric sets (defined in Figure 2) and test thermograms of healthy individuals with a reference set of control thermograms in the temperature interval 59–89 °C.

Calorimetric Group	T_m_^HSA^ (°C)	T_m_^Igs^ (°C)	c_P_^HSA^/c_P_^Igs^	T_FM_ (°C)	ΔH_cal_ (cal/g)	r	P	ρ
control	61.9 ± 0.7	68.4 ± 0.3	2.45 ± 0.72	64.0 ± 1.1	3.1 ± 0.3	0.97 ± 0.20	0.76 ± 0.05	0.80 ± 0.10
MM1	63.1 ± 0.7	68.8 ± 1.4	1.41 ± 0.23	67.0 ± 1.1	3.4 ± 0.2	0.89 ± 0.80	0.71 ± 0.09	0.75 ± 0.09
MM2	66.2 ± 0.9	72.1 ± 0.4	1.38 ± 0.29	68.5 ± 0.9	3.5 ± 0.3	0.66 ± 0.08	0.63 ± 0.08	0.64 ± 0.08
MM3	62.6 ± 1.1	69.3 ± 0.4	0.60 ± 0.13	68.8 ± 0.4	3.8 ± 0.5	0.64 ± 0.08	0.59 ± 0.10	0.61 ± 0.10
MM4	63.0 ± 0.7	72.8 ± 1.4	0.63 ± 0.02	70.1 ± 0.9	4.0 ± 0.1	0.49 ± 0.10	0.59 ± 0.07	0.56 ± 0.08
IgG1	60.0 ± 0.7	75.3 ± 0.4	0.39 ± 0.05	73.3 ± 1.2	3.21 ± 0.9	0.29 ± 0.10	0.42 ± 0.06	0.38 ± 0.06
IgG2	60.8 ± 0.6	67.4 ± 0.2	0.25 ± 0.05	67.3 ± 0.6	4.25 ± 1.1	0.29 ± 0.10	0.56 ± 0.05	0.51 ± 0.07
FLC1	62.9 ± 0.6	69.5 ± 1.1	1.47 ± 0.50	65.7 ± 1.3	3.80 ± 0.50	0.94 ± 0.02	0.72 ± 0.09	0.74 ± 0.02
FLC2	68.2 ± 0.9	77.1 ± 1.3	0.99 ± 0.26	67.8 ± 1.4	3.60 ± 0.60	0.38 ± 0.09	0.55 ± 0.08	0.49 ± 0.07
κFLC-case	60.5	69.1	0.55	67.2	4.43	0.47	0.57	0.54

**Table 3 cancers-14-03884-t003:** Degrees of similarities μ(ci,cj) and differentiation ν(ci,cj) determined for the pairs of criteria ci and cj by means of ICA analyses on a dataset generated for secretory MM cases. The interrelations between the following criteria are presented: excess heat capacity of the immunoglobulins and albumin assigned transitions, c_P_^Igs^ and c_P_^HSA^, respectively, concentration of albumin ([HSA]) and M-proteins ([M]), weighted average center of the thermogram, T_FM_, and similarity metric, ρ.

ci,cj	μ(ci,cj)	ν(ci,cj)
c_P_^Igs^, [M]	0.790	0.163
c_P_^Igs^, c_P_^HSA^	0.239	0.666
c_P_^HSA^, [HSA]	0.560	0.391
c_P_^Igs^, T_FM_	0.713	0.239
c_P_^Igs^, ρ	0.204	0.725

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
