# Peer review of "Calorimetric Markers for Detection and Monitoring of Multiple Myeloma"

_cancers, 2022, doi:10.3390/cancers14163884_

Round 1

Reviewer 1 Report

This review is a compact summary of the group experiments headed by Prof. Taneva from the field of multiple myeloma performed by DSC with a wide outlook on the international publications.

The quality of the instrument is at top level, and during the data evaluation they use the international standards of statistical methods. The achieved results are compared with the thermal parameters of separated plasma compounds. The conclusions have strong clinical relevances.

The construction of paper fit to the standards of Cancers, the text is well written and very easy to read and follow it. I have only some remarks in connection with the typing of calorimetric parameters:

-          In the figures the subscripts and superscripts are written properly but in the text, figure as well as table captions are wrong [e.g.: Tm, and instead of Cp is written cP (it is heat capacity and not specific heat!) and so on]. These should be corrected.

-          During the evaluation they calculate excess heat capacity (my opinion is, that this parameter is less informative for the medical doctors than the calorimetric enthalpy). They express it in cal/gK. The recommendation of ICTAC (this international organisation made an official recommendation for the units of thermal analytical parameters) in connection with this unit is kJ/kgK!

After this minor revision I can warmly propose the publication of this paper.

Author Response

Reviewer 1: This review is a compact summary of the group experiments headed by Prof. Taneva from the field of multiple myeloma performed by DSC with a wide outlook on the international publications.

The quality of the instrument is at top level, and during the data evaluation they use the international standards of statistical methods. The achieved results are compared with the thermal parameters of separated plasma compounds. The conclusions have strong clinical relevances.

The construction of paper fit to the standards of Cancers, the text is well written and very easy to read and follow it.

I have only some remarks in connection with the typing of calorimetric parameters:

In the figures the subscripts and superscripts are written properly but in the text, figure as well as table captions are wrong [e.g.: Tm, and instead of Cp is written cP (it is heat capacity and not specific heat!) and so on]. These should be corrected.

We thank the reviewer for the positive evaluation of our manuscript.

Unfortunately, we didn’t notice that the subscripts and superscripts were not correct, they were probably changed when the text was placed in the template.

We made the necessary corrections in the revised MS.

During the evaluation they calculate excess heat capacity (my opinion is, that this parameter is less informative for the medical doctors than the calorimetric enthalpy). They express it in cal/gK. The recommendation of ICTAC (this international organisation made an official recommendation for the units of thermal analytical parameters) in connection with this unit is kJ/kgK!

We actually used both terms „excess heat capacity” and „amplitude” of the transitions, the latter one is easily understandable. In order to use calorimetric enthalpy instead of heat capacity all of the thermograms (more than 600) have to be deconvoluted. Since the enthalpy of certian transition should be proportional to the heat capacity, we believe that this additional estimation will not provide much new information that would affect the take-home message of the manuscript.

If the reviewer agrees, we prefer not to change the units in this paper because cal/gK was used in most of our previous publications, and it is widely used in the literature relevant to this review, although as the reviewer stated according to the ICTAC the recommended unit is kJ/kgK. Converting the units would make the comparison of data published in different articles much more difficult.

Reviewer 2 Report

Krumova S et al presented in this paper a review about the potential clinical use of differential scanning calorimetry technique applied to multiple myeloma diagnosis and monitoring of its treatment. They described the different studies up to date and mathematical methods applied to this matter that are now available. I considered that the manuscript is clear, well written and it could be useful for someone that wants to go deeper in this matter.

Some major/minor points in the manuscript that could be revised:

- Line 19: macroglobulinemia instead of “macroglobulinemia”.

- Line 68: “the 51 °C peak is assigned to fibrinogen” this is for the case of plasma. In Figure 1 serum is represented so, this peak does not appear. This could be explained for better understanding.

- Line 78: “for a number of diseases [1,4,9,11–18]”, there is one reference regarding to multiple sclerosis that could be added: “Annesi, F.; Hermoso-Durán, S.; Rizzuti, B.; Bruno, R.; Pirritano, D.; Petrone, A.; Del Giudice, F.; Ojeda, J.; Vega, S.; Sanchez-Gracia, O.; Velazquez-Campoy, A.; Abian, O.; Guzzi, R. Thermal Liquid Biopsy (TLB) of Blood Plasma as a Potential Tool to Help in the Early Diagnosis of Multiple Sclerosis. J. Pers. Med. 2021, 11, 295. https://doi.org/10.3390/jpm11040295”.

- Line 79: “for various types of cancer:”, there is one reference regarding to pancreatic cancer diagnosis that could be added: “Hermoso-Durán, S.; García-Rayado, G.; Ceballos-Laita, L.; Sostres, C.; Vega, S.; Millastre, J.; Sánchez-Gracia, O.; Ojeda, J.L.; Lanas, Á.; Velázquez-Campoy, A.; Abian, O. Thermal Liquid Biopsy (TLB) Focused on Benign and Premalignant Pancreatic Cyst Diagnosis. J. Pers. Med. 2021, 11, 25. https://doi.org/10.3390/jpm11010025”.

- Line 79: “for various types of cancer:”, there is one reference regarding to lung cancer diagnosis that could be added: “Rodrigo, A.; Ojeda, J.L.; Vega, S.; Sanchez-Gracia, O.; Lanas, A.; Isla, D.; Velazquez-Campoy, A.; Abian, O. Thermal Liquid Biopsy (TLB): A Predictive Score Derived from Serum Thermograms as a Clinical Tool for Screening Lung Cancer Patients. Cancers 2019, 11, 1012. https://doi.org/10.3390/cancers11071012”

- Table 1: The information included in “statistical measures” row is repeated also in the text. I would refer to the table and comment/explain the parameter meaning. In case that there are reference values for the variables defined in the table, please include it for better understanding.

- Figure 2: Perhaps it would be better if it appears before line 158 for better understanding. Also, information about the number of samples represented could be included.

- Figure 3: If it is positioned beside Figure 2, the bars and the curves could be better contrasted.

- In Table 2 and Figure 4, there is only information about some of the groups in Figures 2 and 3. If there is information about them, it could be included in the supplementary material section (in case that this section is present in the review article).

- A summarizing figure at the end of the paper explaining how DSC could be specifically relevant in clinical diagnosis and treatment monitoring in this specific disease would be nice to be included.

- Line 293, please include the number of patients included in the study that appear in Figure 5.

- In Figure 5, the control group how close to the IgA MM patients is? I mean, are they patients also suffering a general cell transplantation process or are healthy patients without any intervention? This should be mentioned or clarified as serum DSC profiles perhaps could be sensible to this type of medical procedures.

- In Line 326, it is not clear for me which correlation gives information that could be relevant for patient clinical monitoring. Please, add a sentence related to this to better understanding.

Author Response

Reviewer 2: Krumova S et al presented in this paper a review about the potential clinical use of differential scanning calorimetry technique applied to multiple myeloma diagnosis and monitoring of its treatment. They described the different studies up to date and mathematical methods applied to this matter that are now available. I considered that the manuscript is clear, well written and it could be useful for someone that wants to go deeper in this matter.

We thank the reviewer for the comments and suggestions.

Some major/minor points in the manuscript that could be revised:

- Line 19: macroglobulinemia instead of “macroglobulinemia”. ?????

This is corrected in the revised version.

- Line 68: “the 51 °C peak is assigned to fibrinogen” this is for the case of plasma. In Figure 1 serum is represented so, this peak does not appear. This could be explained for better understanding.

The reviewer is right. We have revised the text accordingly (rows 76-77 in the revised MS).

- Line 78: “for a number of diseases [1,4,9,11–18]”, there is one reference regarding to multiple sclerosis that could be added: “Annesi, F.; Hermoso-Durán, S.; Rizzuti, B.; Bruno, R.; Pirritano, D.; Petrone, A.; Del Giudice, F.; Ojeda, J.; Vega, S.; Sanchez-Gracia, O.; Velazquez-Campoy, A.; Abian, O.; Guzzi, R. Thermal Liquid Biopsy (TLB) of Blood Plasma as a Potential Tool to Help in the Early Diagnosis of Multiple Sclerosis. J. Pers. Med. 2021, 11, 295. https://doi.org/10.3390/jpm11040295”.

- Line 79: “for various types of cancer:”, there is one reference regarding to pancreatic cancer diagnosis that could be added: “Hermoso-Durán, S.; García-Rayado, G.; Ceballos-Laita, L.; Sostres, C.; Vega, S.; Millastre, J.; Sánchez-Gracia, O.; Ojeda, J.L.; Lanas, Á.; Velázquez-Campoy, A.; Abian, O. Thermal Liquid Biopsy (TLB) Focused on Benign and Premalignant Pancreatic Cyst Diagnosis. J. Pers. Med. 2021, 11, 25. https://doi.org/10.3390/jpm11010025”.

- Line 79: “for various types of cancer:”, there is one reference regarding to lung cancer diagnosis that could be added: “Rodrigo, A.; Ojeda, J.L.; Vega, S.; Sanchez-Gracia, O.; Lanas, A.; Isla, D.; Velazquez-Campoy, A.; Abian, O. Thermal Liquid Biopsy (TLB): A Predictive Score Derived from Serum Thermograms as a Clinical Tool for Screening Lung Cancer Patients. Cancers 2019, 11, 1012. https://doi.org/10.3390/cancers11071012

Thank you for the suggestion - these references are included in the revised version (19,24,32).

Table 1: The information included in “statistical measures” row is repeated also in the text. I would refer to the table and comment/explain the parameter meaning. In case that there are reference values for the variables defined in the table, please include it for better understanding.

We have omitted the explanatory text from the Table in order to exclude repetition of information. As it is explained in the text “Values close to 1 indicate high similarity of the test thermogram and the control set, while values deviating from 1 are due to statistically significant differences between the two curves.” (row 97-100).

- Figure 2: Perhaps it would be better if it appears before line 158 for better understanding. Also, information about the number of samples represented could be included.

Due to page limitation it is not suitable to place Fig.2 before line 158. The number of studied cases for each MM isotype is now provided in the description of Fig. 2.

- Figure 3: If it is positioned beside Figure 2, the bars and the curves could be better contrasted.

We have revised Figure 3 according the requirements of the other reviewers. We believe in the present form it is more clear.

- In Table 2 and Figure 4, there is only information about some of the groups in Figures 2 and 3. If there is information about them, it could be included in the supplementary material section (in case that this section is present in the review article).

We have included the missing data in Table 2. Unfortunately, we do not have AFM data on all of the measured blood sera. We utilized this technique only for characterization of FLC2 group and 1 unusual FLC case, as mentioned in the text. Therefore, we are not able to provide further data.

- A summarizing figure at the end of the paper explaining how DSC could be specifically relevant in clinical diagnosis and treatment monitoring in this specific disease would be nice to be included.

We have included a schematic representation (Figure 6) of interdisciplinary approach for MM diagnosis involving DSC as a complementary tool to clinical tests.

- Line 293, please include the number of patients included in the study that appear in Figure 5.

We clarified in the figure description that the presented data are derived for “2 patients monitored for 5 or 9 months”. As described in the text those are selected examples of monitoring series of DSC scans recorded during the course of patients treatment, published in refs 40 and 43 (rows 331-340).

- In Figure 5, the control group how close to the IgA MM patients is? I mean, are they patients also suffering a general cell transplantation process or are healthy patients without any intervention? This should be mentioned or clarified as serum DSC profiles perhaps could be sensible to this type of medical procedures.

In the description of Fig. 5 we clarify that the control DSC profile is derived for healthy individuals without any medical intervention.

- In Line 326, it is not clear for me which correlation gives information that could be relevant for patient clinical monitoring. Please, add a sentence related to this to better understanding.

We provided additional information on this point (row 366-374):

“In our earlier work we have determined the specificity and sensitivity of the immunological test (that includes the M-protein concentration and κ/λ FLC ratio) and the calorimetric approach (taking into account the r and TFM parameters) for MM diagnosis [40]. We have found that M-protein level is more sensitive and specific than r and TFM, however for cases when no M-protein is secreted the calorimetric markers perform better than the immunological marker κ/λ FLC ratio. The combination of immunological markers and TFM increased the specificity and sensitivity of disease detection, most pronounced for NS MM cases (for more details see [40]). Therefore, it is our belief that DSC might be useful complementary monitoring tool for MM patients.”

Reviewer 3 Report

This is a review article devoted to the application of differential scanning calorimetry (DSC) for examination of the denaturational heat capacity profiles of blood serum isolated from patients diagnosed with multiple mieloma (MM). The authors have summarized here their observations in this field published in the last years. They have processed an impressive amount of 634 previously recorded by them denaturational profiles of blood sera and have been able to distribute these profiles in distinctly different groups, typified by different transition shapes, possibly corresponding to different types of multiple myeloma. The classification presented in the article exposes the remarkably large differences between the shapes of the MM profiles between themselves as well as with the control thermogram. This appears to be the main result of the present study and it also demonstrates, as the authors have pointed out, the high potential of DSC in studies of multiple myeloma.

Several comments are due.

1) First of all, in view of the very large differences in the fine structures of the MM thermograms with respect to the structure of the control thermogram, it should be possible to sort out the MM profiles into different groups just by visual inspection, without any mathematical processing. It would appear that statistical criteria like similarity, etc., might even be misleading in some cases because they may have the same averaged value for profiles with completely different fine structures. It remains unclear to what extent the profile assignments to different groups were made by the authors by visual inspection and to what extent on basis of the application of the statistical approach. That should be clearly stated in the text. The authors should also comment on what is the usefulness and what additional information is provided by the statistical processing of the DSC profiles. In fact, all the major discrepancies between MM and control profiles pointed out in the Conclusions, such as low- and high-temperature peaks, albumin peak depression (also present in various other diseases), are readily distinguishable with simple eye and do not appear to need statistical criteria for their verification.

2) The authors should add references (for example, to the legend of Fig. 2) to make clear what articles the respective profiles have been taken from.

3) Fig. 3 needs to be reworked, especially with respect to the group placements along the vertical axis and the bar groupings. The legend to Fig. 3 is not sufficient and needs to be expanded. It might be worth presenting the data in Fig. 3 in a table.

4) It would be useful to explain what method did the authors use to determine the baseline positions in the enthalpy calculations. It is well known that protein denaturations proceed with different Cp slopes below and above the transitions. These slopes are rarely horizontal and may pose difficulties in baseline determinations.

Author Response

Reviewer 3: This is a review article devoted to the application of differential scanning calorimetry (DSC) for examination of the denaturational heat capacity profiles of blood serum isolated from patients diagnosed with multiple mieloma (MM). The authors have summarized here their observations in this field published in the last years. They have processed an impressive amount of 634 previously recorded by them denaturational profiles of blood sera and have been able to distribute these profiles in distinctly different groups, typified by different transition shapes, possibly corresponding to different types of multiple myeloma. The classification presented in the article exposes the remarkably large differences between the shapes of the MM profiles between themselves as well as with the control thermogram. This appears to be the main result of the present study and it also demonstrates, as the authors have pointed out, the high potential of DSC in studies of multiple myeloma.

Several comments are due.

1) First of all, in view of the very large differences in the fine structures of the MM thermograms with respect to the structure of the control thermogram, it should be possible to sort out the MM profiles into different groups just by visual inspection, without any mathematical processing. It would appear that statistical criteria like similarity, etc., might even be misleading in some cases because they may have the same averaged value for profiles with completely different fine structures. It remains unclear to what extent the profile assignments to different groups were made by the authors by visual inspection and to what extent on basis of the application of the statistical approach. That should be clearly stated in the text.

We thank the reviewer for the critical evaluation of our manuscript, which helped us to improve our work.

Although the structure of the MM thermograms strongly differs from the healthy ones, MM thermograms were not sorted by visual inspection. Each thermogram was analyzed, and the number, position and amplitude of the resolved transitions, the weighted average center (TFM) and the total enthalpy were determined. In our previous publications, the MM thermograms for the different myeloma isotypes (IgG, IgM, IgA, FLC and NS MM) were grouped on the basis of all the above-mentioned parameters as discussed in our already published results (cited in the MS).

The paragraph on p. 4, rows 134-138 is rewritten in the revised MS and explains how the MM thermograms of the entire database are grouped:

“We reevaluated a database of 643 DSC profiles of MM patients which were either previously grouped in sets of thermograms for each MM type [5, 35-40] or newly recorded. Here, the thermograms are combined in four groups irrespective of the different myeloma types (IgG, IgM, IgA, FLC and NS MM) based on the number of the thermal transitions, the position of the main transition and the similarity in shape of the DSC profiles (Figure 2 a,b).”

The reviewer is right that “the statistical criteria like similarity, etc., might be misleading in some cases because they may have the same averaged value for profiles with completely different fine structures”. However, this might be true if the only criteria used to compare thermograms is statistical one; we have found that the similarity measure for example is a good criterion to combine with the other thermodynamic parameters.

The authors should also comment on what is the usefulness and what additional information is provided by the statistical processing of the DSC profiles. In fact, all the major discrepancies between MM and control profiles pointed out in the Conclusions, such as low- and high-temperature peaks, albumin peak depression (also present in various other diseases), are readily distinguishable with simple eye and do not appear to need statistical criteria for their verification.

Table 2 and the last two paragraphs of “2. Multiple myeloma discrimination and calorimetry-based classification” are dedicated to the explanation of the usefulness of the mathematical approaches for data analysis.

We believe that the statistical evaluation is helpful in order to refine the objective quantification of the differences between the groups and in fact excludes thermograms assignment based on visual inspection.

Therefore, we and others apply different mathematical/statistical approaches with the idea to develop algorithm that will help to assign a newly measured thermogram to predefined specific sets of thermograms associated with a certain disease. For this certain association limits need to be established such as the range of variation of peak position and height. The addition of statistical parameters (like a range of variation of r, P, ρ) would refine those assignments. Algorithm based fast and objective data evaluation is of outmost importance for the implication of DSC in the clinical practice.

In the revised version we added additional information on the usefulness of the applied by us mathematical approach (p. 4, rows 112-113): “This approach revealed correlations between the calorimetric and biochemical parameters of blood plasma/serum which helped for deciphering the nature of the shifts in the thermal stability of the major serum proteins.”

The authors should add references (for example, to the legend of Fig. 2) to make clear what articles the respective profiles have been taken from.

We thank the reviewer for this comment. We have added the relevant references in the Figure legend.

3) Fig. 3 needs to be reworked, especially with respect to the group placements along the vertical axis and the bar groupings. The legend to Fig. 3 is not sufficient and needs to be expanded. It might be worth presenting the data in Fig. 3 in a table.

In the revised version Fig. 3 is modified and we believe it is now more clear and easier to comprehend.

4) It would be useful to explain what method did the authors use to determine the baseline positions in the enthalpy calculations. It is well known that protein denaturations proceed with different Cp slopes below and above the transitions. These slopes are rarely horizontal and may pose difficulties in baseline determinations.

Before analysis buffer-buffer scans, routinely measured before each experiment, were subtracted from the serum scans; the resulting traces were then corrected either by linear baseline or by interpolating sigmoidal baseline (when the slopes of the pre- and post- transition baseline differed) and normalized to the total protein concentration.

This paragraph is included in the revised MS – p.2, rows 58-61.